# Impact of Low Skeletal Muscle Mass on Long-Term Outcomes in Hepatocellular Carcinoma Treated with Trans-Arterial Radioembolization: A Retrospective Multi-Center Study

**DOI:** 10.3390/cancers15215195

**Published:** 2023-10-28

**Authors:** Heechul Nam, Hyun Yang, Ho Soo Chun, Han Ah Lee, Joon Yeul Nam, Jeong Won Jang, Yeon Seok Seo, Do Young Kim, Yoon Jun Kim, Si Hyun Bae

**Affiliations:** 1Department of Internal Medicine, College of Medicine, The Catholic University of Korea, Seoul 06591, Republic of Korea; hcnam128@catholic.ac.kr (H.N.); oneggu@naver.com (H.Y.); garden@catholic.ac.kr (J.W.J.); 2Liver Research Center, The Catholic University, Seoul 06591, Republic of Korea; 3Departments of Internal Medicine, College of Medicine, Ewha Womans University, Seoul 07985, Republic of Korea; lakesy@yuhs.ac (H.S.C.); amelia86@naver.com (H.A.L.); 4Department of Internal Medicine and Liver Research Institute, College of Medicine, Seoul National University, Seoul 03080, Republic of Korea; moreno777@gmail.com (J.Y.N.); yoonjun@snu.ac.kr (Y.J.K.); 5Department of Internal Medicine, College of Medicine, Korea University, Anam Hospital, Seoul 02841, Republic of Korea; drseo@korea.ac.kr; 6Department of Internal Medicine, Yonsei Liver Center, College of Medicine, Yonsei University, Severance Hospital, Seoul 03772, Republic of Korea; dyk1025@yuhs.ac

**Keywords:** hepatocellular carcinoma, trans-arterial radioembolization, Yttrium-90, locoregional treatment, portal vein tumor thrombosis, overall survival, progression-free survival, skeletal muscle mass, body mass index

## Abstract

**Simple Summary:**

Trans-arterial radioembolization (TARE) can effectively treat patients with locally advanced hepatocellular carcinoma (HCC). However, TARE is not widely used in clinical practice due to the lack of consensus on how to select the best candidates for this therapy. Therefore, it is crucial to find predictive biomarkers or clinical parameters that can differentiate the patients who will respond well to TARE and improve their outcomes. Low skeletal muscle mass (LSMM) is one of the factors that affects the prognosis of HCC patients and has been extensively studied in relation to different treatments. We used a semi-automated program that has been validated and proven to be reliable for assessing LSMM. In this study, we aimed to identify prognostic factors, including pre-treatment LSMM, for survival outcomes in patients with HCC undergoing TARE.

**Abstract:**

Trans-arterial radioembolization (TARE) is a form of radiation therapy performed for hepatocellular carcinoma (HCC) via selective intra-arterial injection of Yttrium-90-loaded microspheres. This was a multi-center retrospective study of consecutive patients with HCC who underwent TARE between July 2009 and May 2019. Using pre-treatment computed tomography imaging, the total cross-sectional area (cm^2^) of the abdominal skeletal muscle at the third lumbar vertebra was measured. The skeletal muscle index (SMI) was calculated by normalizing the muscle area to patient height. In total, 347 patients (median age, 65 years; 284 male) were included in the study. A total of 108 (31.1%) patients had portal vein tumor thrombus (PVTT), and 126 (36.3%) were classified as LSMM. The median overall survival (OS) was 28.1 months (95% CI, 24.8–35.7), and median progression-free survival was 8.0 months (95% CI, 6.4–9.4). Multivariate Cox regression analysis revealed that LSMM (hazard ratio [HR], 1.36; 95% CI, 1.00–1.85, *p* = 0.05), PVTT (HR, 1.82; 95% CI, 1.33–2.49, *p* < 0.01), alpha-fetoprotein (AFP) (≥200 ng/mL) (HR 1.41; 95% CI, 1.04–1.92, *p* = 0.03), and albumin–bilirubin grade (2–3) (HR 1.74; 95% CI, 1.24–2.43, *p* < 0.01) were independently associated with poor OS. TARE provided favorable long-term outcomes for patients with advanced HCC. Pre-treatment LSMM independently associated with survival, suggesting its utility as a surrogate biomarker for identifying TARE candidates.

## 1. Introduction

Hepatocellular carcinoma (HCC) is a major cause of cancer-related deaths worldwide, and its incidence is steadily increasing [1]. Despite progress in the treatment and surveillance of HCC, its incidence and mortality rates continue to increase [2]. Early detection of HCC is important to reduce mortality by providing more opportunities for curative treatment. However, many patients are diagnosed with HCC at an intermediate or advanced stage, when curative treatment is not feasible. Most clinical practice guidelines recommend locoregional treatment (LRT) for patients with HCC who are ineligible for curative treatment [3,4,5]. Trans-arterial chemoembolization (TACE) is the most commonly used LRT worldwide. Despite its widespread use, TACE has several drawbacks, including the need for multiple treatments, variable effectiveness, and difficulties in implementation in advanced diseases with vascular invasion.

Trans-arterial radioembolization (TARE) is a type of radiation therapy that involves the selective injection of microspheres containing Yttrium-90 (^90^Y) into hepatic arteries. ^90^Y is a pure beta emitter that enables the delivery of high radiation doses to tumor cells, while preserving adjacent hepatic tissues [6]. Additionally, ^90^Y treatment utilizes small-sized (20–60 μm) microspheres, which have minimal impact on hepatic arterial blood flow, thus allowing for its administration to patients with portal vein invasion. Given the aforementioned advantages, TARE is becoming an increasingly promising LRT option that can overcome the drawbacks of TACE.

The 2018 American Association for the Study of Liver Diseases guidelines for HCC stated that TARE can be used to treat patients at various stages of HCC [4]. In early-stage HCC, TARE can be curative, whereas in the intermediate to advanced stages, it can serve as a bridge or downstaging treatment option. However, owing to the diverse stages of patients receiving TARE, treatment outcomes and prognoses can vary widely and are difficult to predict.

Sarcopenia is characterized by the progressive loss of skeletal muscle mass and function associated with aging or chronic diseases. Sarcopenia has been shown to be associated with unfavorable outcomes in various cancers, including HCC [7,8,9,10]. It can negatively impact liver function, increase inflammation throughout the body, decrease tolerance to treatments, and lower the quality of life in patients with HCC. This updated consensus paper on sarcopenia focuses on low muscle strength as a key characteristic of sarcopenia, uses detection of low muscle quantity and quality to confirm the sarcopenia diagnosis, and identifies poor physical performance as indicative of severe sarcopenia [11]. Computed tomography (CT)-detected low skeletal muscle mass (LSMM) has been identified as a predictor of poor prognosis in HCC patients, as demonstrated in previous meta-analyses [12,13]. While CT-based measurements are limited to quantifying muscle quantity and cannot assess muscle quality, they have been well established as a significant prognostic factor in HCC. However, the impact of LSMM on the survival and tumor response in patients undergoing TARE remains unclear.

This study aimed to identify prognostic factors in patients with HCC who underwent TARE and to evaluate the impact of pre-treatment LSMM on treatment response, overall survival (OS), and progression-free survival (PFS).

## 2. Materials and Methods

### 2.1. Study Population

This retrospective, multi-center, observational study included consecutive patients with HCC who underwent TARE at five referral centers in Korea. The study was conducted in accordance with the 1975 Declaration of Helsinki and was approved by the local Ethics Committees of each participating center. The study included all adult patients from each participating center who were treated with TARE between July 2009 and May 2019 and underwent an abdominal CT scan capturing L3 within 1 month of treatment. The exclusion criteria for the study included a previous history of systemic chemotherapy for HCC, extrahepatic metastasis, an Eastern Cooperative Oncology Group performance status (ECOG PS) greater than 1, or a pathologically confirmed diagnosis of combined hepatocellular–cholangiocarcinoma.

### 2.2. Study Endpoints

The primary endpoint of this study was OS, which was measured from the start of treatment until death from any cause. The secondary endpoints included PFS and treatment response at 3 months after treatment. The treatment response was assessed according to the modified Response Evaluation Criteria in Solid Tumors (mRECIST) [14].

### 2.3. Data Collection

The clinical characteristics of the patients were obtained through a retrospective review of their medical records at the initiation. These characteristics encompassed demographic information, anthropometric measurements, ECOG PS, laboratory findings (liver function test, alpha-fetoprotein [AFP] and protein induced by vitamin K absence or antagonist-II [PIVKA-II]), and radiologic findings (number and size of intrahepatic lesions and macrovascular invasion). Additionally, the Barcelona Clinic Liver Cancer (BCLC) stage and hepatic function at baseline were obtained using the Child–Pugh classification or albumin–bilirubin (ALBI) grade. The ALBI score was calculated using serum albumin and total bilirubin values with the formula: ALBI score = (log10 bilirubin (µmol/L) × 0.66) + (albumin (g/L) × − 0.085). The dates of disease progression, death, or last follow-up were also collected.

### 2.4. TARE Procedure

TARE was performed using either ^90^Y resin microspheres (SIR-Spheres; Sirtex Medical Ltd., Sydney, Australia) or ^90^Y glass microspheres (TheraSphere; BTG International Ltd., Ottawa, ON, Canada), using standardized techniques. Standardized techniques encompass a set of established and well-documented procedures for the administration of ^90^Y resin microspheres and ^90^Y glass microspheres. Dose calculation of administered activity was based on body surface area (resin) or partition model (glass) [6]. The procedure was administered 10–14 days after a preprocedural examination, which included an evaluation of the lung shunt fraction and 99mTc scintigraphy (planar or SPECT) to measure the lung shunt fraction. For TheraSphere, a lung dose fraction of >30 Gy per treatment (or a total of 50 Gy for all estimated infusions) was the limit. For SIR-Spheres, a 20% lung shunt was the limit, with reduced activity recommended for patients with a 10–20% lung shunt.

### 2.5. Outcomes and Assessments

In all centers, patients were followed up using radiological studies, including CT or magnetic resonance imaging. To evaluate the treatment response, radiological studies were performed 3 months after TARE. Laboratory tests, including liver function and tumor marker tests, were regularly measured during subsequent follow-up visits. The frequency and type of follow-up tests were determined based on each patient’s individual case and treatment plans. In cases where patients did not respond to TARE, alternative therapeutic approaches were implemented and tailored to the individual patient’s circumstances.

### 2.6. Measurement of Skeletal Muscle Mass

We utilized a semi-automated open-source software (BMI measurement tools, version 1.0; https://sourceforge.net/projects/muscle-fat-area-measurement (accessed on 30 June 2020)) to measure the cross-sectional area (cm^2^) of the skeletal muscles based on attenuation segmentation. The Hounsfield unit (HU) thresholds ranged from −29 to 150 HU for skeletal muscle mass, −190 to −30 HU for subcutaneous fat, and −150 to −50 HU for visceral fat, as shown in Figure 1. Measurements were obtained from CT scans performed within 1 month prior to TARE at the level of the transverse processes of the third lumbar (L3) vertebra. The cross-sectional area of the muscle was segmented by a single trained researcher (HY), who was blinded to subject identification and outcomes. Through drawing a single peritoneal line on the CT image, researchers could easily and swiftly obtain measurements, including abdominal circumference, abdominal muscle area, subcutaneous fat area, and visceral fat area.

### 2.7. Definition of LSMM

The cross-sectional muscle area was normalized by height squared (in meters) to calculate the skeletal muscle index (SMI) (cm^2^/m^2^). We applied optimal stratification to determine the body mass index (BMI) and sex-specific cutoffs for SMI to identify LSMM [7]. For men, the SMI cut-offs were <43 cm^2^/m^2^ for underweight or normal-weight patients (BMI < 25 kg/m^2^) and <53 cm^2^/m^2^ for overweight (BMI ≥ 25 kg/m^2^) or obese patients (BMI ≥ 30 kg/m^2^). For women, the cut-off value was <41 cm^2^/m^2^, regardless of weight.

### 2.8. Statistical Analysis

Categorical variables are presented as counts and percentages [n (%)]. Continuous variables are expressed as either the mean ± standard deviation or median with interquartile range and were transformed into two-level categorical data based on their median values. Appropriate statistical tests, including the Chi-square test, Fisher’s exact test, Student’s *t*-test, and Mann–Whitney U-test, were used for the analysis. The Kaplan–Meier method was used to estimate the cumulative incidence of events over time. Differences between groups were analyzed using the log-rank test, and the Bonferroni correction was applied to account for multiple comparisons. Univariate and multivariate Cox proportional hazards regression models were used to assess the risk factors for OS and PFS. Statistical significance was set at *p* < 0.05. All statistical analyses were performed using STATA software 17 (StataCorp LLC, College Station, TX, USA).

## 3. Results

### 3.1. Baseline Characteristics

A total of 347 patients who underwent TARE were included in this study. The baseline patient characteristics are presented in Table 1. The median age of the patients was 65 years, 81.8% were male patients, and the mean BMI was 23.9 kg/m^2^. Hepatitis B virus (HBV) infection was the major cause of liver disease (57.3%). Nearly all patients (94.5%) were classified as Child–Pugh class A, and almost half (42.9%) were classified as ALBI grade 1. The median tumor diameter (largest) was 8.4 (6.2–10.6) cm, and 53.6% of patients had multifocal tumors. Portal vein tumor thrombus (PVTT) was found in 46.0% of the patients. According to BCLC staging, 18.7% of the patients had BCLC stage A, 41.5% had BCLC stage B, and 39.8% had BCLC stage C. The mean follow-up time was 28.6 months ± 25 (95% confidence interval [CI]; 4.3–112.2).

The patients were divided into no-LSMM (n = 221) and LSMM (n = 126) groups. The baseline characteristics revealed significant differences between the two groups in terms of LSMM. The LSMM group was older, had a lower BMI, and had a higher proportion of female patients and patients with alcoholic liver disease than the no-LSMM group. Albumin levels also showed significant differences between the two groups, resulting in a better ALBI grade in the no-LSMM group. However, tumor characteristics, such as median tumor diameter, proportion of multifocal tumors, presence of PVTT, and tumor markers (AFP and PIVKA-II), did not show significant differences between the two groups.

### 3.2. Response to TARE

During the study, none of the patients underwent repeated TARE of the same target lesion, even in cases of local tumor progression. The median administered radiation activity was 2.5 GBq, and the median hepatopulmonary shunt was 6.7%. Additional detailed information on the TARE procedure is provided in Appendix A. Response evaluations 3 months after TARE were available for 333 patients, representing 96.0% of the study population. According to the mRECIST criteria, complete response, partial response, and stable disease were observed in 26 (7.8%), 121 (36.3%), and 106 (31.8%) patients, respectively. The overall response rate was 44.1%, and the disease control rate was 76.0% (Table 2). The no-LSMM group demonstrated significantly higher objective response (*p* = 0.031) and disease control (*p* < 0.001) rates than the LSMM group.

### 3.3. Factors Associated with Overall Survival

The median OS was 28.1 (95% CI, 24.8–35.7) months, and the 1-, 3-, and 5-year OS rates were 74.9% (95% CI, 69.8–79.2%), 43.5% (95% CI, 37.2–49.6%), and 37.2% (95% CI, 30.7–43.6%), respectively (Figure 2A). The no-LSMM group had a significantly higher OS than the LSMM group, with a median OS of 35.3 (95% CI, 27.1–56.9) months and 21.1 (95% CI, 14.8–27.6) months, respectively (*p* = 0.002) (Figure 2B). The OS was significantly higher in patients without PVTT than in those with PVTT, with a median OS of 37.7 (95% CI, 29.4–63.3) months and 13.0 (95% CI, 9.6–22.0) months, respectively (*p* < 0.001) (Figure 2C). LSMM, PVTT, AFP level (≥200 ng/mL), ALBI grade (2–3), tumor number (≥2), and largest tumor diameter (>8 cm) were identified as significant factors in univariate analysis (Table 3). Multivariate analysis revealed that LSMM (hazard ratio [HR], 1.36; 95% CI, 1.00–1.85; *p* = 0.048), PVTT (HR, 1.82; 95% CI, 1.33–2.49; *p* < 0.001), AFP (≥200 ng/mL) (HR 1.41; 95% CI, 1.04–1.92; *p* = 0.028), and ALBI grade (2–3) (HR 1.74; 95% CI, 1.24–2.43; *p* = 0.001) were independently associated with OS.

### 3.4. Factors Associated with Progression-Free Survival

The median PFS was 8.0 (95% CI, 6.4–9.4) months, and the 1-, 3-, and 5-year PFS rates were 38.9% (95% CI, 33.7–44.2%), 20.4% (95% CI, 15.8–25.4%), and 13.1% (95% CI, 8.9–18.1%), respectively (Figure 3A). Analysis of factors affecting PFS yielded findings consistent with the aforementioned OS analysis results. The no-LSMM group demonstrated significantly better outcomes than the LSMM group (*p* = 0.002) (Figure 3B), and the patients without PVTT had significantly better outcomes than those with PVTT (*p* < 0.001) (Figure 3C). LSMM (HR 1.54; 95% CI, 1.19–1.98; *p* = 0.001), PVTT (HR 2.08; 95% CI, 1.59–2.72; *p* < 0.001), AFP (≥200 ng/mL) (HR 1.38; 95% CI, 1.07–1.77; *p* = 0.013), ALBI grade (2–3) (HR 1.49; 95% CI, 1.15–1.93; *p* = 0.002), and tumor number (≥2) (HR 1.55; 95% CI, 1.20–2.01; *p* = 0.001) were independently correlated with PFS (Table 3).

## 4. Discussion

TARE is a safe and effective treatment for patients with locally advanced HCC who are not amenable to curative-intent treatments, such as surgical resection or ablative therapies, or for those who do not respond to other therapies. TACE is the most commonly used LRT for HCC; however, it often requires repeated procedures, and the rate of TACE refractoriness is considerable [15]. Some institutions have adopted TARE as the first-line trans-arterial LRT for HCC [16]. However, the widespread clinical application of TARE is often constrained by its comparatively high cost in comparison to other LRTs, the limited availability of facilities and expertise, and the absence of a consensus on patient selection criteria. Concerns about the high cost and the potential for radioembolization-induced liver injury make it difficult to administer repeat treatments in some cases. Therefore, the identification of predictive biomarkers or clinical characteristics that can identify patients likely to respond to TARE would be invaluable for optimizing treatment outcomes.

In this study, we included 65 patients with HCC and BCLC stage A. It is not a standard practice to administer TARE treatment to BCLC stage A patients. In our study, BCLC A patients who underwent TARE exhibited distinctive characteristics, notably, a relatively large tumor size with a median of 6.7 cm (interquartile range (IQR); 4.8–9.3) and an advanced age, with a median of 74 years old (IQR; 67–80). In many cases, older patients with larger tumor sizes often face greater challenges in undergoing curative treatments, including surgical resection. TARE can be considered a significant treatment option for patients with these characteristics. TARE demonstrated favorable long-term outcomes for large (≥5 cm), single-nodular HCC without PVTT when compared to surgical resection [17]. Additionally, Salem et al. reported a study that demonstrated the long-term preservation of health-related quality of life in HCC patients undergoing TARE [18]. Therefore, we have decided to employ TARE in elderly patients with substantial tumor sizes, as alternative treatment options may pose considerable challenges for this particular patient group. Additionally, we used two types of microspheres with distinct sizes and properties, which could potentially lead to differences in their effects, including the embolic effect. However, in prior studies that investigated the differences between resin and glass microspheres, these two types did not exhibit significant disparities in outcomes [19].

Sarcopenia and cancer cachexia are complex syndromes characterized by a gradual loss of skeletal muscle mass and are widely recognized as predictors of poor survival in various types of cancer [7]. Sarcopenia can result in decreased functional reserves and reduced ability to carry out daily activities. Additionally, it is often accompanied by conditions such as insulin resistance, vitamin D deficiency, and increased levels of inflammatory cytokines, all of which are associated with the progression of liver fibrosis and HCC [8]. Consequently, sarcopenia can significantly affect the prognosis of HCC, and several studies have explored its association with patients undergoing various treatments [20,21,22]. Often devoid of functional parameters, the predominant approach to defining sarcopenia in malignancies, including HCC, relies on the assessment of CT images and the measurement of muscle indices. This method primarily focuses on LSMM and is considered a convenient and readily applicable approach in routine cancer staging [12]. Accurate assessment of muscle mass is crucial to identify patients who are at risk. To achieve this, it is important to use the most reliable and consistent method among the various methods available for measuring muscle mass. CT attenuation-based segmentation is a commonly used method for distinguishing fat from other soft tissues, and various software programs have been used to quantify body composition. This study utilized a semi-automated program that has been validated in multiple studies and demonstrated high reliability [23,24,25].

This study investigated the association between LSMM and the long-term outcomes of TARE in patients with HCC. Although the exact mechanisms underlying this relationship are not fully understood, one possible explanation is that the immune response has been shown to have a significant impact on TARE outcomes. Valerie et al. reported a correlation between immune activation and sustained response to TARE [26]. Their study found that the robust activation of CD8+ T cells within both the tumor and the systemic environment was significantly associated with positive TARE outcomes. Given that sarcopenia is closely linked to host immunity, this could provide an insight into our findings. Chronic inflammation in cancer can contribute to the development of sarcopenia, partly due to immune dysfunction, such as T-cell exhaustion [27]. Skeletal muscle tissue produces myokines, such as interleukin (IL)-6, IL-15, tumor necrosis factor-α, and transforming growth factor-β, and altered myokine activity can induce immune senescence in sarcopenia. Therefore, patients with LSMM who receive TARE may have a poor prognosis.

Compared to TACE or sorafenib, TARE has various advantages as a treatment for locally advanced HCC, in terms of quality of life, treatment tolerability, and time to progression [18,28,29]. Despite these advantages, TARE did not demonstrate improved survival compared to sorafenib as a first-line treatment for unresectable HCC patients in recent randomized controlled trials [30,31]. However, a more recent randomized trial (DOSISPHERE-01) showed significantly improved OS with a personalized dosimetry model compared to a standard-dose calculation model [32]. The unique features of TARE, including its intricate relationship with immunity and the requirement for personalized dosimetry models, underscore the importance of refining patient selection and individualizing treatment applications to enhance survival outcomes. According to our findings, no LSMM, no PVTT, an ALBI grade of 1, and low AFP levels (<200 ng/mL) were independent predictors of favorable OS following TARE. These findings highlight the importance of patient stratification and personalized treatment for optimizing treatment outcomes.

PVTT is a notable prognostic factor for TARE. Unlike TACE, TARE is considered a relatively safe and effective LRT that can be used even in cases accompanied by PVTT. Some studies have reported the therapeutic efficacy of TARE in patients with PVTT, and a predictive model for outcomes has been developed for this population [16,33]. However, our results indicate that both OS and PFS were significantly superior in patients without PVTT. Although TARE can still be used in cases with PVTT, it appears to show better outcomes in patients without PVTT. Along these lines, some studies have shown that TARE has comparable effects to surgery in treating large HCC without PVTT and can be applied as a potential curative therapy for early-stage HCC [17,34]. There is an emerging opinion that TARE should be used in the early or intermediate stages rather than in the advanced stages based on such evidence [35]. The findings of our study corroborate this perspective, as we observed superior long-term OS and PFS in patients without PVTT. However, further investigation is required to fully understand the implications of these results.

This study had several limitations that should be acknowledged. First, the lack of a validation set may have affected the generalizability and reliability of our findings. Nevertheless, our study is one of the largest retrospective studies to date that investigate the impact of LSMM on long-term outcomes of TARE. Obtaining a significant number of patients who underwent TARE can be challenging, as it requires experienced medical professionals and is relatively costly. Future studies should include a validation set to further corroborate our results. Second, this study was conducted in a single country and in patients with the same ethnicity. Korea is an HBV-endemic area with a relatively small proportion of obese patients. Therefore, the results of our study may not be applicable to populations with different etiologies and body composition. In particular, further validation is required in other countries, where NASH-associated HCC constitutes a significant portion of cases. Third, many patients underwent additional treatments after TARE that may have influenced their prognoses. However, LSMM affects the efficacy and tolerability of subsequent treatments and long-term prognosis, underscoring its importance. Another limitation of this study was that cross-sectional muscle mass measurements were conducted by a single clinical physician, and there were no validation measurement results available. Nevertheless, it is important to emphasize that the semi-automated program used by the authors is a validated measurement method known for its high intraclass correlation coefficient [36]. Lastly, we only evaluated muscle quantity and did not investigate its functional aspects. The updated consensus paper on sarcopenia by the European Working Group on Sarcopenia in Older People 2 places a strong emphasis on low muscle strength as a key characteristic of sarcopenia and uses the detection of low muscle quantity and quality to confirm the diagnosis of sarcopenia [11]. Given the overall absence of functional parameters in this study, we have chosen to use the term ‘low skeletal muscle mass’ instead of ‘sarcopenia’.

## 5. Conclusions

In conclusion, our analysis demonstrated that TARE is an effective therapeutic option for patients with advanced HCC. Notably, pre-treatment LSMM is identified as an independent prognostic factor for both long-term OS and PFS. Additionally, PVTT, ALBI grade, and the level of AFP were significant prognostic factors for outcomes. These findings underscore the importance of incorporating LSMM as a useful biomarker in implementing TARE. To optimize survival outcomes of TARE, further studies are warranted to refine patient selection and personalize treatment strategies.

## Figures and Tables

**Figure 1 cancers-15-05195-f001:**
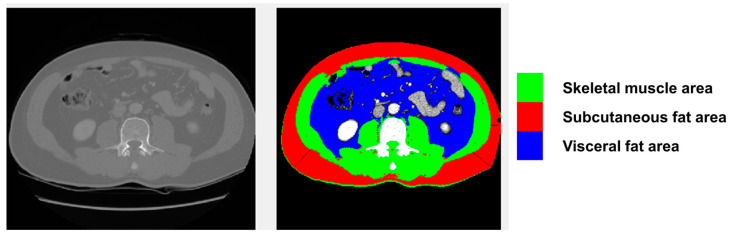
Assessment of body composition using body mass index measurement software, showing the proportions of skeletal muscle (green), subcutaneous fat (red), and visceral fat (blue) in a representative patient.

**Figure 2 cancers-15-05195-f002:**
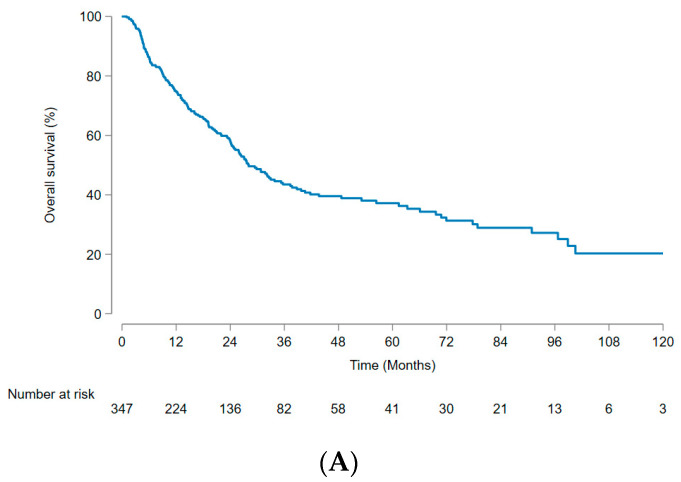
Kaplan–Meier curve for overall survival. (**A**) Total patients; (**B**) subgroup analysis stratified according to LSMM; (**C**) subgroup analysis stratified according to PVTT. LSMM, low skeletal muscle mass; PVTT, portal vein tumor thrombus.

**Figure 3 cancers-15-05195-f003:**
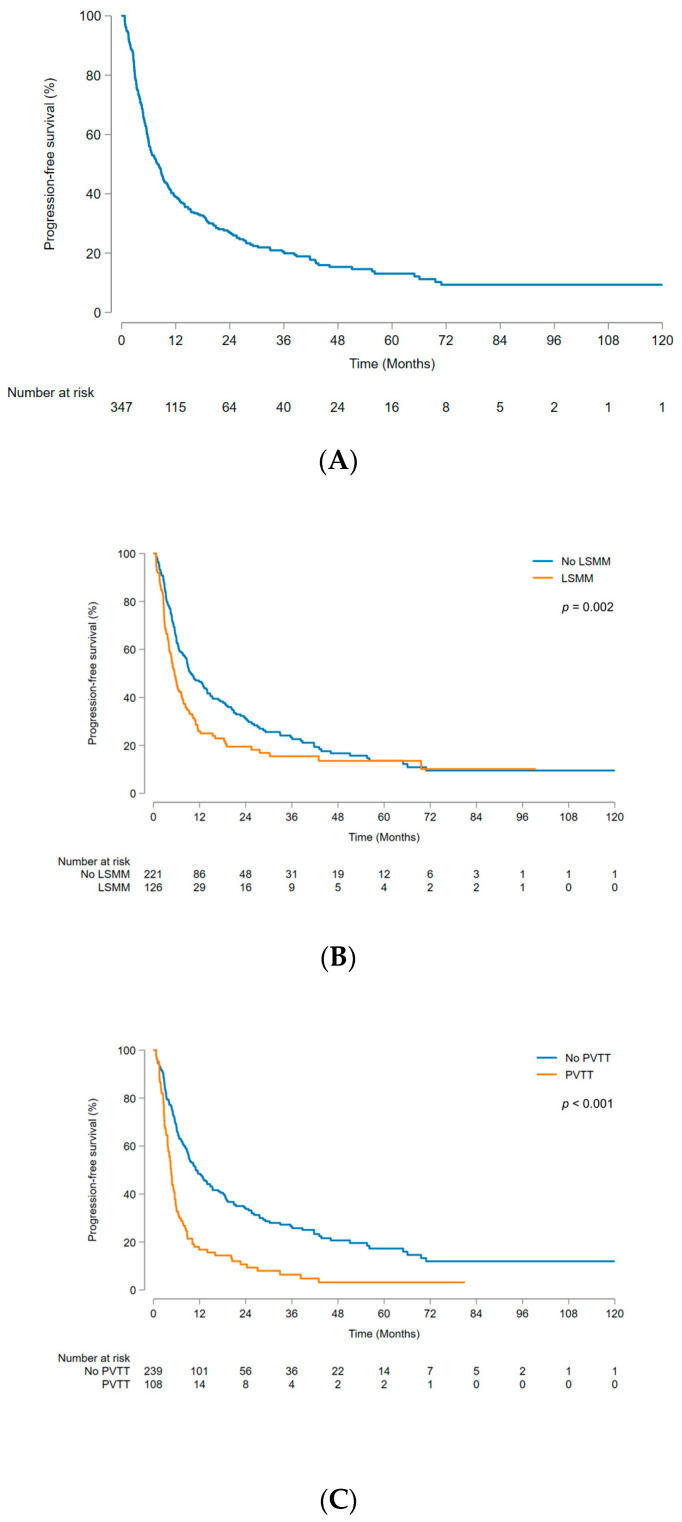
Kaplan–Meier curve for progression-free survival. (**A**) Total patients; (**B**) subgroup analysis stratified according to LSMM; (**C**) subgroup analysis stratified according to PVTT. LSMM, low skeletal muscle mass; PVTT, portal vein tumor thrombus.

**Table 1 cancers-15-05195-t001:** Baseline characteristics of the patients treated with TARE.

Variables	Total (n = 347)	No LSMM (n = 221)	LSMM (n = 126)	*p* Value
Median age, years	65 (55–75)	62 (54–72)	71 (59–78)	<0.001
Male sex	284 (81.8%)	195 (88.2%)	89 (70.6%)	<0.001
Mean BMI (Kg/m^2^)	23.9 (3.4)	24.5 (3.2)	22.9 (3.4)	<0.001
Etiology of disease *				0.010
Hepatitis B virus	199 (57.3%)	141 (63.8%)	58 (46.0%)	
Hepatitis C virus	35 (10.1%)	17 (7.7%)	18 (14.3%)	
Alcohol	59 (17.0%)	29 (13.1%)	30 (23.8%)	
Non-alcoholic steatohepatitis	35 (10.1%)	22 (10.0%)	13 (10.3%)	
Other	19 (5.5%)	12 (5.4%)	7 (5.6%)	
ECOG PS				0.560
0	289 (83.3%)	186 (84.2%)	103 (81.7%)	
1	58 (16.7%)	35 (15.8%)	23 (18.3%)	
Mean total bilirubin (mg/dL)	0.8 (0.6)	0.8 (0.6)	0.7 (0.6)	0.063
Mean albumin (g/dL)	3.7 (0.5)	3.8 (0.5)	3.6 (0.5)	0.002
Median INR	1.1 (1.0–1.1)	1.1 (1.0–1.1)	1.0 (1.0–1.2)	0.200
Child–Pugh class				0.130
A	328 (94.5%)	212 (95.9%)	116 (92.1%)	
B	19 (5.5%)	9 (4.1%)	10 (7.9%)	
ALBI grade				0.023
1	149 (42.9%)	107 (48.4%)	42 (33.3%)	
2	190 (54.8%)	109 (49.3%)	81 (64.3%)	
3	8 (2.3%)	5 (2.3%)	3 (2.4%)	
Median tumor diameter (cm)	8.4 (6.2–10.6)	8.0 (6.1–10.0)	9.0 (6.5–11.6)	0.059
Tumor number (≥2)	186 (53.6%)	119 (53.8%)	67 (53.2%)	0.900
PVTT	108 (31.1%)	73 (33.0%)	35 (27.8%)	0.310
BCLC stage				0.860
A	65 (18.7%)	40 (18.1%)	25 (19.8%)	
B	144 (41.5%)	94 (42.5%)	50 (39.7%)	
C	138 (39.8%)	87 (39.4%)	51 (40.5%)	
Median AFP (ng/mL)	103.0 (10.7–2235.3)	58.1 (11.7–1958.5)	198.1 (9.6–2922.0)	0.240
Median PIVKA-II (mAU/mL)	961.0 (108.0–7032.0)	862.0 (96.0–5592.0)	1740.0 (132.5–13607.0)	0.120

Variables expressed as mean (standard deviation), median (interquartile range), or n (%). LSMM, low skeletal muscle mass; BMI, body mass index; ECOG PS, Eastern Cooperative Oncology Group performance status; INR, international normalized ratio; ALBI grade, albumin–bilirubin grade; TACE, trans-arterial chemoembolization; PVTT, portal vein tumor thrombus; BCLC, Barcelona Clinic Liver Cancer; AFP, alpha-fetoprotein; PIVKA-II, prothrombin induced by vitamin K absence or antagonist-II. * The same patient could have several causes of disease.

**Table 2 cancers-15-05195-t002:** Treatment responses after 3 months of treatment among evaluable patients.

Variables	Total (n = 333)	no LSMM (n = 212)	LSMM (n = 121)	*p* Value
Overall response				0.007
Complete response	26 (7.8%)	18 (8.5%)	8 (6.6%)	
Partial response	121 (36.3%)	85 (40.1%)	36 (29.8%)	
Stable disease	106 (31.8%)	71 (33.5%)	35 (28.9%)	
Progressive disease	80 (24.0%)	38 (17.9%)	42 (34.7%)	
Objective response	147 (44.1%)	103 (48.6%)	44 (36.4%)	0.031
Disease control rate	253 (76.0%)	174 (82.1%)	79 (65.3%)	<0.001

Data are n (%). LSMM, low skeletal muscle mass.

**Table 3 cancers-15-05195-t003:** Univariate and multivariate analysis of factors influencing OS and PFS.

Characteristics	Overall Survival			Progression-Free Survival		
	Univariate Analysis	Multivariate Analysis		Univariate Analysis	Multivariate Analysis	
	*p* Value	HR (95% CI)	*p* Value	*p* Value	HR (95% CI)	*p* Value
Age (≥65)	0.555			0.208		
Sex (male)	0.168			0.518		
Etiology (viral hepatitis)	0.931			0.250		
LSMM (yes)	0.002	1.363 (1.002–1.854)	0.048	0.002	1.535 (1.192–1.978)	0.001
PVTT (yes)	<0.001	1.822 (1.333–2.491)	<0.001	<0.001	2.075 (1.585–2.716)	<0.001
AFP (≥200 ng/mL)	<0.001	1.411 (1.037–1.920)	0.028	<0.001	1.376 (1.070–1.770)	0.013
ALBI (grade 2–3)	<0.001	1.737 (1.244–2.425)	0.001	<0.001	1.492 (1.152–1.934)	0.002
Tumor number (≥2)	0.001	1.304 (0.946–1.798)	0.105	<0.001	1.553 (1.198–2.014)	0.001
Largest tumor diameter (>8 cm)	0.001	1.348 (0.986–1.843)	0.061	0.012	1.056 (0.819–1.362)	0.674
BMI (>25 kg/m^2^)	0.316			0.248		
Previous TACE (yes)	0.068			0.140		

OS, overall survival; PFS, progression-free survival; LSMM, low skeletal muscle mass; PVTT, portal vein tumor thrombus; AFP, alpha-fetoprotein; BMI, body mass index; TACE, trans-arterial chemoembolization.

## Data Availability

The original contributions presented in the study are included in the article/Appendix A; further inquiries can be directed to the corresponding author.

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
