# Peer review of "Impact of Low Skeletal Muscle Mass on Long-Term Outcomes in Hepatocellular Carcinoma Treated with Trans-Arterial Radioembolization: A Retrospective Multi-Center Study"

_cancers, 2023, doi:10.3390/cancers15215195_

Round 1
Reviewer 1 Report
Comments and Suggestions for Authors
Cruz-Jentoft AJ, Bahat G, Bauer J, Boirie Y, Bruyère O, Cederholm T, Cooper C, Landi F, Rolland Y, Sayer AA, Schneider SM, Sieber CC, Topinkova E, Vandewoude M, Visser M, Zamboni M; Writing Group for the European Working Group on Sarcopenia in Older People 2 (EWGSOP2), and the Extended Group for EWGSOP2. Sarcopenia: revised European consensus on definition and diagnosis. Age Ageing. 2019 Jan 1;48(1):16-31. doi: 10.1093/ageing/afy169. Erratum in: Age Ageing. 2019 Jul 1;48(4):601. PMID: 30312372; PMCID: PMC6322506.
" In this updated consensus paper on sarcopenia, EWGSOP2: (1) focuses on low muscle strength as a key characteristic of sarcopenia, uses detection of low muscle quantity and quality to confirm the sarcopenia diagnosis, and identifies poor physical performance as indicative of severe sarcopenia; (2) updates the clinical algorithm that can be used for sarcopenia case-finding, diagnosis and confirmation, and severity determination and (3) provides clear cut-off points for measurements of variables that identify and characterise sarcopenia"
Congratulations for your article, very interesting overall, but I am sorry to note that the definition of "Sarcopenia" that you have given is not in accordance with the most recent trends in scientific literature, so I advise you to modify the term "sarcopenia" with " muscle mass" .
Author Response
In accordance with Academic editor`s feedback, we acknowledge with regret that our study solely evaluated muscle quantity, without delving into its functional aspects. Unfortunately, due to the retrospective nature of this research, we encounter substantial constraints in incorporating additional assessments of muscle strength. We sincerely regret these limitations.
We have revised the term "sarcopenia" to "low skeletal muscle mass (LSMM)" in both the title and the main text. Furthermore, we have incorporated the following information into the "Limitations" section
“Lastly, we only evaluated muscle quantity and did not investigate its functional aspects. The updated consensus paper on sarcopenia by the European Working Group on Sarco-penia in Older People 2 places a strong emphasis on low muscle strength as a key charac-teristic of sarcopenia and uses the detection of low muscle quantity and quality to confirm the diagnosis of sarcopenia.[11] Given the overall absence of functional parameters in this study, we have chosen to use the term ' low skeletal muscle mass' instead of 'sarcopenia’.”
Additionally, we have provided explanations regarding the relationship between sarcopenia and LSMM in both the introduction and discussion sections.

Reviewer 2 Report
Comments and Suggestions for Authors
I wish to congratulate the authors on their study. I would like to know why the study included 65 patients with HCC and BCLC stage A, as they should either be resected or studied for a liver transplant.
I agree with the authors and believe this study needs external validation with different populations. Occidental countries have a significant obesity problem, which has led to an increase in NASH (non-alcoholic steatohepatitis) cases and more HCC cases.
Author Response
Reviewer #2: Comments and Suggestions for Authors
We kindly thank the reviewer #2 for his or her helpful insights. We address specific points below:
Comment 1
I wish to congratulate the authors on their study. I would like to know why the study included 65 patients with HCC and BCLC stage A, as they should either be resected or studied for a liver transplant.
Response 1
In line with the reviewer's observation, it is indeed uncommon to administer TARE treatment to BCLC stage A patients as a standard practice. In our study, BCLC A patients underwent TARE exhibited distinctive characteristics, notably a relatively large tumor size with a median of 6.7cm (interquartile range; 4.8 - 9.3) and an advanced age, with a median of 74 years old (interquartile range; 67 - 80). In many cases, older patients with larger tumor sizes, as observed in our study's BCLC stage A cohort, often face greater challenges in undergoing curative treatments, including surgical interventions, compared to the typical BCLC stage A patients.
There are several studies suggest that TARE can be considered a significant treatment option for patients with these characteristics. We would like to kindly introduce the reviewer to the study that demonstrated favorable long-term outcomes of TARE for large, single nodular HCC compared to surgical resection [Kim et al. J Nucl Med 2022, 63, 1215-1222]. According to the study, in patients with large (≥5 cm), single nodular HCC without portal vein tumor thrombosis, TARE demonstrated comparable overall survival (adjusted HR, 1.04; 95% CI, 0.42-2.59; P = 0.93) and time to progression (adjusted HR, 0.98; 95% CI, 0.50-1.95; P = 0.96) while maintaining a superior safety profile when compared with surgical resection. Additionally, Salem et al. reported a study that demonstrated the long-term preservation of health-related quality of life in HCC patients undergoing TARE [Clin Gastroenterol Hepatol 2013, 11, 1358-1365 e1351]. With this context in mind, we have made the decision to employ TARE in elderly patients with substantial tumor sizes, as alternative treatment options may pose considerable challenges for this particular patient group.
To enhance clarity, I have added the following content to the Discussion section.
“In this study, we included 65 patients with HCC and BCLC stage A. It is not a standard practice to administer TARE treatment to BCLC stage A patients. In our study, BCLC A patients underwent TARE exhibited distinctive characteristics, notably a relatively large tumor size with a median of 6.7cm (interquartile range (IQR); 4.8 - 9.3) and an advanced age, with a median of 74 years old (IQR; 67 - 80). In many cases, older patients with larger tumor sizes, often face greater challenges in undergoing curative treatments, including surgical resection. TARE can be considered a significant treatment option for patients with these characteristics. TARE demonstrated favorable long-term outcomes for large (≥5 cm), single nodular HCC without PVTT when compared to surgical resection. Additionally, Salem et al. reported a study that demonstrated the long-term preservation of health-related quality of life in HCC patients undergoing TARE. Therefore, we have decided to employ TARE in elderly patients with substantial tumor sizes, as alternative treatment options may pose considerable challenges for this particular patient group.”
Comment 2
I agree with the authors and believe this study needs external validation with different populations. Occidental countries have a significant obesity problem, which has led to an increase in NASH (non-alcoholic steatohepatitis) cases and more HCC cases.
Response 2
I wholeheartedly agree with your perspective. While this is a multi-institutional study, it's essential to acknowledge that our analysis was conducted exclusively on the Korean population, making generalization a challenging task and one of the limitations of our study.
I have added the following content to the Discussion section.
“In particular, further validation is required in other countries, where NASH-associated HCC constitutes a significant portion of cases.”

Reviewer 3 Report
Comments and Suggestions for Authors
Summary
In their work, the authors assess sarcopenia as possible prognostic factor for TARE with 90Y. The population includes 347 patients from different centers.
The manuscript is well-written and clear. The results are clearly presented.
General Comments
I suggest to the authors to assess whether the devices used (either resin or glass spheres) results in a statistically significant difference in the results. I would add this analysis only if statistically significant.
Introduction
Line 63/64/65: The authors state: Additionally, Y90 treatment utilizes small-sized (20-35 μm) microspheres, which have minimal impact on hepatic arterial blood flow, thus allowing for its administration to patients with portal vein invasion. However, according to EANM guidelines (DOI 10.1007/s00259-011-1812-2) glass spheres have a diameter between 20 and 30 μm while resin spheres between 20 and 60 μm.
a. Could the authors clarify this in the text?
b. Could the author also add the spheres density and its embolic effect?
c. Do the author believe that resin spheres, being bigger and with a moderate embolic effect, impact the blood flow? Please add this point to the discussion.
Materials and Methods
Line 116: The authors state:TARE was performed using either Y90 resin microspheres (SIR-Spheres; Sirtex Medical Ltd., Sydney, Australia) or Y90 glass microspheres (TheraSphere; BTG International 115 Ltd., Ottawa, Canada) using standardized techniques.
a. Could the authors specify what do they mean by standardized techniques? Do they also refer to the method used to determine the administered activity?
b. Given that resin spheres use the BSA method to determine the administered activity, do the author believe that this method, being indirectly linked to sarcopenia, can have influenced the treatment outcome (e.g. sarcopenia will lead to lower BSA, thus to lower injected activity and eventually lower dose to tumors)?
Lines 121-124: The authors refer to SPECT and partition model. However it is a bit unclear why they do refer to these. I would suggest to remove the sentence and:
a. Specify which image technique (SPECT or planar) has been used to compute the lung shunt fraction
b. Specify which method (BSA/partition model/other) has been used to compute the administered activity.
Lines 141-143: The authors state that a single researcher performed the segmentation of the tissues. However they prior report threshold values based on which the semi-automatic segmentation was performed.
a. Can the authors specify what is manually required for the semi automatic segmentation?
b. Could the author add the single researcher as a study limitation in the discussion session?
Results
Table 2. Can the authors specify what the p-value of the overall survival (0.007) refers to?
Discussion
Line 256 - 266: The authors state: The incapability to administer repeat treatment for recurrent or progressive disease. Could the authors specify the guidelines they refer to? In Europe the repeat treatment of recurrent disease is possible, but the guidelines might differ based on region.
As mentioned above, I would include in the discussion the impact of the different sphere devices and the method used to define the administered activity
Specific Comments
1. Figure 1, I would suggest to either add the color as an overlay with some transparency or to also show the original CT. This would leave to the reader the possibility to visually assess the accuracy of the semi-automatic segmentation method used.
2. I would refer to yttrium-90 as 90Y throughout the text, instead of Y90. However, I will leave this to the authors, based on the guidelines for the volume the manuscript has been submitted to
3. For both Figure 2 and Figure 3, I would suggest to use number between 0 and 100 for the y scale, since the y label is expressed as percentage.
4. In the supplemental material there are two minor typos: hepatopulmonart instead of hepatopulmonary and, in the caption, radioembolizati instead of radioembolization. Please check these.
5. I would specify in the table which variables are expressed by mean and which by median.
Author Response
Reviewer #3:
Thank you very much for taking the time to review this manuscript. Please find the detailed responses below and the corresponding revisions/corrections highlighted/in track changes in the re-submitted files.
Comments 1
I suggest to the authors to assess whether the devices used (either resin or glass spheres) results in a statistically significant difference in the results. I would add this analysis only if statistically significant.
Response 1
We thank the reviewer for the comment. We have indeed conducted an analysis to assess the influence of resin or glass spheres on the outcomes. However, as the p-values were 0.6989 for OS and 0.4538 for PFS, indicating no statistically significant associations, we have appropriately refrained from including these results in the main text.
Comments 2
Introduction
Line 63/64/65: The authors state: Additionally, Y90 treatment utilizes small-sized (20-35 μm) microspheres, which have minimal impact on hepatic arterial blood flow, thus allowing for its administration to patients with portal vein invasion. However, according to EANM guidelines (DOI 10.1007/s00259-011-1812-2) glass spheres have a diameter between 20 and 30 μm while resin spheres between 20 and 60 μm.
a. Could the authors clarify this in the text?
b. Could the author also add the spheres density and its embolic effect?
c. Do the author believe that resin spheres, being bigger and with a moderate embolic effect, impact the blood flow? Please add this point to the discussion.
Response 2
We thank the reviewer for the valuable comment.
- I made an error during the process of entering precise numerical values. I have corrected the main text as follows.
“Aditionally, Y90 treatment utilizes small-sized (20-60 μm) microspheres, -”
- As the reviewer pointed out, there are indeed notable differences between glass-based and resin-based devices in terms of microsphere composition, size, degree of embolic effect, and specific activity per sphere. Glass-based 90Y therapy is typically regarded as having a minimal embolic effect, while resin-based 90Y therapy has been described as having a moderately embolic effect. These distinctions are significant in understanding the characteristics and effects of the two therapies.
In the study by Biederman et al., which compared resin-based 90Y and glass-based 90Y microspheres in TARE treatment [J Vasc Interv Radiol. 2016 Jun;27(6):812-821.e2], the authors observed that the treatment response in patients with HCC and PVT did not show a significant difference between the 90Y -glass and 90Y -resin groups. Treatment outcomes of TARE cannot be attributed solely to the embolic effect. Additionally, the existing literature that directly compares the embolic effect of 90Y -resin and 90Y -glass is limited, which poses challenges in establishing clear distinctions in this regard.
- As mentioned earlier, given the larger size of resin spheres, one might expect a greater embolic effect. However, similar to the findings of the prior study by Biederman et al., our analysis results also suggest that this factor does not seem to significantly impact clinical outcomes of TARE. The primary endpoint of this study is not to compare the differences between the two types of microspheres, and therefore, we did not collect data to assess their impact on embolic effect and blood flow. Additionally, it's important to consider that the retrospective nature of the study makes it challenging to conduct additional data analysis at this stage.
I have added the following content to the Discussion section.
“We used two types of microspheres with distinct sizes and properties, which could potentially lead to differences in their effects, including the embolic effect. However, in prior studies that investigated the differences between resin and glass microspheres, these two types did not exhibit significant disparities in outcomes.”
Comments 3
Materials and Methods
Line 116: The authors state:TARE was performed using either Y90 resin microspheres (SIR-Spheres; Sirtex Medical Ltd., Sydney, Australia) or Y90 glass microspheres (TheraSphere; BTG International 115 Ltd., Ottawa, Canada) using standardized techniques.
a. Could the authors specify what do they mean by standardized techniques? Do they also refer to the method used to determine the administered activity?
b. Given that resin spheres use the BSA method to determine the administered activity, do the author believe that this method, being indirectly linked to sarcopenia, can have influenced the treatment outcome (e.g. sarcopenia will lead to lower BSA, thus to lower injected activity and eventually lower dose to tumors)?
Response 3
- In reference to your query about the term "standardized techniques," we appreciate your diligence in seeking clarification. In our manuscript, "standardized techniques" encompass a set of established and well-documented procedures for the administration of Y90 resin microspheres and Y90 glass microspheres. This involves various aspects of the treatment process, including patient preparation, catheterization techniques, and microsphere delivery protocols. And, dose calculation of administered activity was based on body surface area (resin) or liver weight partition model (glass).
The specific details of these administered activity calculations have been included in our methodology section.
“Standardized techniques encompass a set of established and well-documented procedures for the administration of 90Y resin microspheres and 90Y glass microspheres. [Choi et al. J Liver Cancer. 2022 Mar;22(1):4-13]. Dose calculation of administered activity was based on body surface area (resin) or partition model (glass).”
- The reviewer indeed raised a crucial question regarding the impact of low BSA on the prognosis and treatment outcomes. To address this, we calculated the median BSA for all patients and further analyzed its influence on outcomes. In our study, the median BSA for patients was 1.73 (IQR 1.62-1.85). Using this median value as a reference, we conducted analyses to assess its impact on OS and PFS. The results showed that BSA did not significantly influence OS (P-value 0.486) or PFS (P-value 0.334). Focusing on the patients treated with 90Y resin microspheres, for whom the administered dose was calculated based on BSA (n=212), BSA still did not exhibit a significant impact on both OS (P-value 0.693) and PFS (P-value 0.219). These results collectively suggest that low BSA may not significantly influence treatment outcomes.
It's important to note that basic anthropometric measurements such as BMI or BSA may not provide an accurate assessment of muscle mass. Therefore, the primary aim of this study was to measure muscle mass and investigate its impact on HCC treatment, given that muscle mass assessment may offer more insights into treatment outcomes.
Comments 4
Lines 121-124: The authors refer to SPECT and partition model. However it is a bit unclear why they do refer to these. I would suggest to remove the sentence and:
a. Specify which image technique (SPECT or planar) has been used to compute the lung shunt fraction
b. Specify which method (BSA/partition model/other) has been used to compute the administered activity.
Response 4
I agree with the reviewer's comment. It caused confusion as an unnecessary expression, so I have removed that sentence.
- We thank the reviewer for bringing this point to our attention. Due to the retrospective nature of this study, which observed long-term outcomes across multiple institutions, uniform methods could not be employed consistently across institutions and time periods. Depending on the patient, both planar and SPECT 99mTc scintigraphy were utilized to measure the lung shunt fraction.
I have added the following content to the Methods section.
“99mTc scintigraphy (planar or SPECT) to measure the lung shunt fraction.”
- We agree with the reviewer's concern that our previous expression caused confusion. We have clarified this point in the methods section.
“Dose calculation of administered activity was based on body surface area (resin) or partition model (glass).”
Comments 5
Lines 141-143: The authors state that a single researcher performed the segmentation of the tissues. However they prior report threshold values based on which the semi-automatic segmentation was performed.
a. Can the authors specify what is manually required for the semi automatic segmentation?
b. Could the author add the single researcher as a study limitation in the discussion session?
Response 5
- The program we utilized operates by allowing researchers to define a virtual line for distinguishing the peritoneum. Once a few reference points are marked through several clicks, the program then automatically measures parameters such as abdominal circumference, abdominal muscle area, subcutaneous fat area, and visceral fat area. The accuracy and clinical significance of this program have been established in multiple studies. [Kang, et al. J Cachexia Sarcopenia Muscle 2018, 9, 860-870.], [Goh, et al. Sci Rep 2022, 12, 7647.], [Kim, et al. J Cachexia Sarcopenia Muscle 2023, 14, 860-868.] I will ensure to provide a more explicit explanation of how to use the program in the manuscript.
We will ensure to provide a clearer explanation of how to use the program in the method as follows.
“Through drawing a single peritoneal line on the CT image, researchers could easily and swiftly obtain measurements, including abdominal circumference, abdominal muscle area, subcutaneous fat area, and visceral fat area.”
- Thank you for the valuable feedback. We will incorporate the following content into the limitation section.
“Another limitation of this study was that cross-sectional muscle mass measurements were conducted by a single clinical physician, and there were no validation measurement results available. Nevertheless, it is important to emphasize that the semi-automated program used by the authors is a validated measurement method known for its high intraclass correlation coefficient [Medicine. 2019 May;98(22):e15867.]”
Comments 6
Results
Table 2. Can the authors specify what the p-value of the overall survival (0.007) refers to?
Response 6
The P-value you mentioned is related to the analysis of the difference in overall `response` between non-sarcopenic and sarcopenic patients. In this study, non-sarcopenic patients demonstrated significantly higher objective response rates and disease control rates, which also reflects in the comparison of overall response rates.
Comments 7
Discussion
Line 256 - 266: The authors state: The incapability to administer repeat treatment for recurrent or progressive disease. Could the authors specify the guidelines they refer to? In Europe the repeat treatment of recurrent disease is possible, but the guidelines might differ based on region.
As mentioned above, I would include in the discussion the impact of the different sphere devices and the method used to define the administered activity
Response 7
Reviewer's insightful comment is greatly appreciated. As you pointed out, repeated treatment is indeed not impossible, and this holds true for the context of Korea where the study was conducted. However, given the high treatment costs, and especially considering that TARE is not reimbursed by the National Health Insurance in Korea, the issue of cost makes repeated treatments challenging. I inadvertently expressed this in a way that could lead to misunderstandings. To make this clearer, I have revised the manuscript as follows.
“Concerns about the high cost and the potential for radioembolization-induced liver injury make it difficult to administer repeat treatments in some cases."
Comments 8
Figure 1, I would suggest to either add the color as an overlay with some transparency or to also show the original CT. This would leave to the reader the possibility to visually assess the accuracy of the semi-automatic segmentation method used.
Response 8
Thank you for your valuable comment. As you pointed out, having the CT images before measurements would enhance readers' understanding. We have submitted the revised figures.
Comments 9
I would refer to yttrium-90 as 90Y throughout the text, instead of Y90. However, I will leave this to the authors, based on the guidelines for the volume the manuscript has been submitted to
Response 9
Following the reviewer's comment and the suggestion for the accurate expression '90Y,' we have revised the manuscript. Thank you for bringing this to our attention.
Comments 10
For both Figure 2 and Figure 3, I would suggest to use number between 0 and 100 for the y scale, since the y label is expressed as percentage.
Response 10
I made an error in how I presented the data. As the reviewer kindly pointed out, we have submitted figures with the y-scale adjusted to '0 and 100.' Thank you for bringing this to our attention.
Comments 11
In the supplemental material there are two minor typos: hepatopulmonart instead of hepatopulmonary and, in the caption, radioembolizati instead of radioembolization. Please check these.
Response 11
Thank you for bringing this to my attention. I appreciate your diligence in addressing the typo, and I'm glad to hear that the corrected version has been uploaded in the supplemental material.
Comments 12
I would specify in the table which variables are expressed by mean and which by median.
Response 12
We have added the mean and median values as suggested in the table footnote.
“Variables expressed as mean (standard deviation)†, median (interquartile range)‡ or n (%).”
